# Metabolite Profiling and Association Analysis of Leaf Tipburn in Heat-Tolerant Bunching Onion Varieties

**DOI:** 10.3390/plants14020187

**Published:** 2025-01-11

**Authors:** Tetsuya Nakajima, Reina Yamamoto, Kanako Matsuse, Masato Fuji, Koei Fujii, Sho Hirata, Mostafa Abdelrahman, Muneo Sato, Masami Yokota Hirai, Masayoshi Shigyo

**Affiliations:** 1Laboratory of Vegetable Crop Science, Division of Life Science, Graduate School of Sciences and Technology for Innovation, Yamaguchi University, Yamaguchi 753-8515, Japan; 2Laboratory of Vegetable Crop Science, Faculty of Agriculture, Yamaguchi University, Yamaguchi 753-8515, Japan; 3Laboratory of Vegetable Crop Science, Division of Yamaguchi University and Kasetsart University Joint Master’s Degree Program in Agricultural and Life Sciences, Yamaguchi University, Yamaguchi 753-8515, Japan; 4Yamaguchi Prefectural Agriculture and Forestry General Technology Center, 10318 Mure, Hofu 747-0004, Japan; 5Laboratory of Agroecology, Department of Bioresource Sciences, Faculty of Agriculture, Kyushu University, 744 Motooka, Nishi-ku, Fukuoka 819-0395, Japan; 6Center of Biotechnology and Genomics, Texas Tech University, Lubbock, TX 49409, USA; 7RIKEN Center for Sustainable Resource Science, 1-7-22 Suehiro-cho, Tsurumi-ku, Yokohama 230-0045, Japan; 8Department of Applied Biosciences, Graduate School of Bioagricultural Sciences, Nagoya University, Nagoya 464-8601, Japan

**Keywords:** *Allium fistulosum*, leaf tipburn, metabolite profiling, organosulfur compound, pigment compounds, functional components

## Abstract

The bunching onion is an important leafy vegetable, prized for its distinctive flavor and color. It is consumed year-round in Japan, where a stable supply is essential. However, in recent years, the challenges posed by climate change and global warming have resulted in adverse effects on bunching onions, including stunted growth, discoloration, and the development of leaf tipburn, threatening both crop quality and yield. Furthermore, as bunching onion belongs to the *Allium* genus, which includes globally significant vegetables such as onion and garlic, studying the impact of climate change on bunching onion serves as an ideal model. The insights gained can also be applied to other crops and regions. This study investigates the effects of different summer growth conditions on the metabolite profile of heat-tolerant bunching onions with dark green leaf blade coloration and examines their association with leaf tipburn. Pigment compound quantification, functional component analysis, leaf tipburn rate assessment, and widely targeted metabolome profiling were performed across two commercial $F_{1}$ varieties, one purebred variety, and six Yamaguchi Prefecture-bred $F_{1}$ lines under different growing conditions. The results obtained were subjected to comparative analyses based on the varieties and groups classified by high and low leaf tipburn rates. The results revealed that β-carotene accumulation peaked with May sowing and July harvest, while the highest accumulation of other pigment compounds was observed with May sowing and September harvest. Additionally, metabolome analysis related to leaf tipburn rates identified several organosulfur compounds, with gamma-glutamyl-propenyl cysteine sulfoxide emerging as one of the key compounds. Based on the intensity data, the fold change of this metabolite was calculated to be 1.66, indicating an increase in the leaf tipburn group compared to the control group. In the control groups, organosulfur compounds appeared to undergo turnover in preparation for stress response. In contrast, in the leaf tipburn groups, it is hypothesized that organosulfur compounds were converted into precursors of pungency, resulting in inadequate responses to stress. This study aims to elucidate the mechanisms through which organosulfur compounds transition into pungent compounds and to develop varieties with improved resistance to leaf tipburn.

## 1. Introduction

The bunching onion (*Allium fistulosum* L.), also known as the welsh onion, green onion, spring onion, or scallion, is widely distributed from Siberia to tropical Asia, particularly in East Asia, where numerous varieties have adapted to diverse local environmental conditions [1]. The distinct flavor and vibrant color of bunching onions contribute to the richness of dishes, leading consumers to expect high-quality bunching onions that offer exceptional flavor, color, and beneficial functional components [2,3]. Bunching onions are consumed year-round in Japan, necessitating a stable and consistent supply to meet market demand. In recent years, producers have sought traits such as “heat tolerance”, to minimize leaf tipburn and poor growth during summer, as well as “dark green coloration” [1,4], and Japanese seed companies have developed summer $F_{1}$ lines characterized by these traits. The chemical composition of bunching onions has been reported to fluctuate across various growth stages and harvest times [5]. Therefore, it is crucial to systematically investigate and understand the variations in pigment compounds and functional components influenced by growing conditions to ensure consistent quality. Furthermore, recent global warming has led to a decline in both production and quality during the summer, with the occurrence of leaf tipburn becoming particularly severe [6].

Tipburn is a physiological disorder that occurs during rapid plant growth and is characterized by necrosis at the leaf apex of young, developing leaves [7]. As bunching onions are leafy vegetables, the deterioration of the appearance of the leaf blade significantly reduces the commercial value, posing a major challenge for farmers and distributors [8,9]. Tipburn is generally considered a calcium-related disorder or a calcium deficiency-related disorder [10,11,12]. The occurrence of tipburn in bunching onions has been reported to be induced by a combination of factors, including soil dryness, high temperatures, prolonged sunlight exposure, and calcium deficiency [13]. Additionally, in onions (*A. cepa*), apart from the effects of pests and diseases, deficiencies in nitrogen, sulfur [14], and boron [15], as well as exposure to ozone [16] and salinity [17], can also cause tipburn. Tipburn in onions is often regarded as a general symptom of stress, as it can be triggered by a wide range of biotic and abiotic stress factors [18]. Leaf tipburn affects not only the *Allium* genus, including crops such as bunching onions, onions, garlic (*A. sativum*), and chives (*A. tuberosum*), but also other crops, such as lettuce *(Lactuca sativa*), white cabbage (*Brassica oleracea* L. var. capitata), Chinese cabbage *(B. pekinensis* (Lour.) Rupr.), Brussels sprouts (*B. oleracea* var. gemmifera DC.), and strawberry (*Fragaria × ananassa* Duch.) [19,20,21]. Therefore, it is crucial to systematically investigate and comprehend the factors contributing to this phenomenon across different species.

Metabolomics serves as a powerful tool for biological discoveries in plants, offering insights into the biochemical status of plant cells at specific developmental stages and in response to environmental conditions [22]. Metabolomics has the potential to greatly enhance the study of stress biology in plants and other organisms by identifying various compounds, including stress metabolism by-products, molecules involved in stress signal transduction, plant hormones, and compounds associated with the plant’s acclimation response [23,24]. In *Allium* species, metabolome analyses have been conducted in various studies focusing on the effects of stress, cultivation conditions, treatment methods, and even the impact of adding alien chromosomes [25,26,27]. Despite extensive research on tipburn in leafy crops such as lettuce (*Lactuca sativa* L.), onion, and cauliflower (*Brassica oleracea* L.), most of these studies have primarily focused on nutritional deficiencies or genotype-related tipburn symptoms [18,28,29]. There is limited knowledge of the biochemical and physiological mechanisms underlying this disorder in bunching onions, particularly under combined environmental stresses. This study addresses this gap by integrating metabolomics with pigment and functional component analyses to uncover the mechanisms of tipburn development and identify strategies for improving tolerance in heat-tolerant $F_{1}$ hybrid and purebred varieties.

## 2. Results

### 2.1. Variation in Pigment Contents and Functional Components Among Different Bunching Onion Varieties and Breeding Lines Across Different Growing Conditions

Images of the plants under each growing condition are shown in the Appendix A. The results for chlorophyll *a*, chlorophyll *b*, lutein, β-carotene, violaxanthin, neoxanthin, total phenolic compounds, total flavonoid compounds, and fructan, as well as the statistical results under various growth conditions, are shown in the Appendix A. The varieties KAMI and NATS showed significantly lower SPAD values, whereas YSG1 and YAM1-5 exhibited significantly higher SPAD values (Figure 1). On the other hand, the NAKA variety showed an intermediate SPAD value trend. Pigment compounds associated with SPAD values, including chlorophyll *a* and *b*, lutein, violaxanthin, and neoxanthin, exhibited trends consistent with the SPAD measurements across the investigated varieties. In contrast, β-carotene exhibited only minor variations among the different varieties (Appendix A). Additionally, functional components, including total phenolic compounds and total flavonoid compounds, were significantly lower in KAMI and significantly higher in YSG1, whereas the other varieties displayed intermediate levels, with no significant differences observed (Appendix A).

PCA and K-means clustering were applied to assess the varietal characteristics under different growing conditions, focusing on samples sown on 7 May and harvested on 18 July (1), 22 August (2), and 18 September (3). The data were projected onto two principal components (PC1 and PC2), which together explained 80.13% of the total variance, with PC1 accounting for 62.10% and PC2 for 18.03% (Figure 2A). The loadings of PC1 were primarily associated with pigment compounds (excluding β-carotene), with an increase in pigment levels on the positive side and a decrease on the negative side. The loadings of PC2 were associated with functional components, including β-carotene, showing increases on the positive side and β-carotene on the negative side (Figure 2B). K-means clustering classified all samples into four clusters: cluster 1 included samples with increased β-carotene, cluster 2 included samples with increased pigment compounds, cluster 3 included samples with decreased pigment compounds, and cluster 4 included samples with increased functional components. Focusing on the variety, NAKA, NATS, and KAMI were grouped into clusters 3 and 4, positioned on the negative side of PC1. In contrast, YAM1-5 was classified into clusters 1, 2, and 4 and was situated on the positive side of PC1 (Table 1). Focusing on the growing conditions, samples from May sowing and July harvest (1) were part of clusters 0 or 2 and were positioned on the negative side of the second principal component, while samples harvested in the following months ((2) and (3)) shifted to the positive side.

Similarly, PCA and K-means clustering were performed using results from May sowing and September harvest (4), June sowing and September harvest (5), and July sowing and September harvest (6). These samples were also projected onto two principal components, accounting for 71.27% of the total variance. PC1 contributed 55.06%, while PC2 contributed 16.21% (Figure 3A). The loadings of PC1 were predominantly associated with pigment compounds, with increased values on the positive side and decreased values on the negative side. PC2 loadings were linked to functional components, including β-carotene, showing an increase on the positive side and a decrease on the negative side (Figure 3B). K-means clustering grouped all samples into four distinct clusters: cluster 1 consisted of samples with elevated levels of both pigment compounds and functional components; cluster 2 included samples with reduced levels of both; cluster 3 contained samples with increased pigment compounds and decreased functional components; cluster 4 was comprised of samples with decreased pigment compounds and increased functional components. YAM1-5 and YSG1 were classified into clusters 1 and 3, while NAKA, NATS, and KAMI were grouped into clusters 2 and 4 (Table 2).

### 2.2. Estimating the Metabolites Involved in Leaf Tipburn

The results for leaf tipburn and metabolome analysis under various growth conditions are presented in the Appendix A. The leaf tipburn rate across 153 samples from 9 varieties ranged from 0.00% to 1.23% under six different growth conditions, with no significant differences observed between varieties (Figure 4). Based on these findings, we selected the top 5% of samples with the highest leaf tipburn rates (eight samples: leaf tipburn group) and the bottom 5% with the lowest rates (eight samples: control group) for further comparative analysis using the 267 metabolites identified from these samples.

To gain deeper insights into the contributions of specific metabolites in the leaf tipburn and control groups, we performed PLS-DA and evaluated the results using VIP scores. The analysis revealed that the leaf tipburn and control groups could be differentiated into two clusters based on two primary components (Figure 5A). Component 1 contributed 11.4%, and Component 2 contributed 17.2%, with a cumulative contribution of 28.6%, though this remains relatively low. The VIP scores highlighted the importance of metabolites, particularly sulfur compounds such as cystathionine, gamma-glutamyl-propenyl-cysteine sulfoxide (gamma-Glu-PRENCSO), and cysteine, as well as flavonoids, in differentiating the groups. Notably, cystathionine and cysteine were found at higher concentrations in the control group (and lower in the leaf tipburn group), while gamma-Glu-PRENCSO was elevated in the leaf tipburn group (and lower in the control group) (Figure 5B). We then conducted a Student’s *t*-test on the 267 metabolites between the leaf tipburn and control groups, followed by FDR correction for *p*-values. No metabolites showed significant differences between the two groups at a significance level of *p* = 0.05. As a result, we selected metabolites with significant uncorrected *p*-values and a $\log_{2}FC$ [leaf tipburn/control] of ≥0.58 or ≤−0.58, equivalent to a 1.5-fold change. This selection identified 11 metabolites (8 increased, 3 decreased) (Figure 6). Many of these metabolites were also highlighted by high VIP scores from the PLS-DA analysis.

Finally, we applied a random forest regression model using the full dataset, with the leaf tipburn rate as the target variable and the 267 metabolites as input features. SHAP analysis was used to interpret the model’s output. The top 20 metabolites ranked by average SHAP values were primarily associated with flavonoids and organosulfur compounds (Figure 7). Among these, gamma-Glu-PRENCSO stood out as a key metabolite, overlapping with those identified by high VIP scores from the PLS-DA analysis and significant changes in the Student’s *t*-test, exhibiting a 1.5-fold change. Additionally, the fold change calculated based on raw intensity data showed a 1.66-fold increase in the leaf tipburn group compared to the control group (Appendix A). Gamma-Glu-PRENCSO is an intermediate of PRENCSO, a precursor of pungency-related compounds. In the leaf tipburn group, organosulfur compounds tended to proceed toward the alliin synthesis pathway rather than the sulfur-assimilation- or antioxidant-related pathways. This metabolic shift may lead to a deficiency in metabolites essential for stress responses, potentially resulting in inadequate adaptation to stress in the leaf tipburn group.

### 2.3. Variations in Organosulfur Compounds and Plant Hormones

Based on the analysis results related to leaf tipburn, several organosulfur compounds associated with the pungency of bunching onion were identified. A heat map was created to compare metabolites related to organosulfur compounds and stress response hormones—ethylene, abscisic acid (ABA), and jasmonic acid (JA)—between the leaf tipburn group and the control group. In the control group, most organosulfur compounds showed an increase, with notable increases in cysteine and cystathionine. Additionally, there was a confirmed decrease in the accumulation of propenyl cysteine sulfoxide (PRENCSO) and gamma-Glu-PRENCSO, which are the main precursors of pungency. Conversely, in the leaf tipburn group, there was a general decrease in organosulfur compounds, while the accumulation of PRENCSO and gamma-Glu-PRENCSO increased. Regarding plant hormones, the precursor to ethylene—1-aminocyclopropane-1-carboxylic acid—increased in the control group, along with ABA and JA. These findings suggest that leaf tipburn is not caused by ethylene synthesis. Furthermore, since JA did not increase in the leaf tipburn group, it is unlikely that leaf tipburn is due to biotic stress, such as disease or herbivore damage (Figure 8A,B).

## 3. Discussion

### 3.1. Clarification of the Differences in Pigment Compounds and Functional Components Among Varieties and Lines Based on the Growing Conditions

We examined the effects of various growing conditions across nine varieties. For plants sown in May and harvested in July, August, and September, β-carotene levels increased in the July harvest (Figure 2A, Table 1 and Appendix A). In the August harvest, β-carotene decreased, while functional components such as phenolic compounds, flavonoids, and fructans increased (Figure 2A, Table 1 and Appendix A). By the September harvest, both functional components and pigment compounds showed increases (Figure 2A, Table 1 and Appendix A). For plants sown in May, June, and July and harvested in September, the earliest sowing time (resulting in the longest growing period) led to a higher accumulation of both pigment compounds and functional components. In contrast, no significant differences were observed in the accumulation of these compounds between the June and July sowing dates (Figure 3, Table 2). In summary, these results indicate that β-carotene content was the highest for May sowing and July harvest, while pigment compounds and functional components accumulated more with May sowing and September harvest. The observed changes in carotenoid content and composition are closely linked to environmental conditions, with light playing a critical role in regulating carotenoid biosynthesis in chloroplasts [30]. For instance, in cucumbers, it has been reported that β-carotene accumulation increases under extended daylight conditions in photoperiod-sensitive varieties [31]. This suggests that the higher β-carotene levels in plants sown in May and harvested in July may be associated with the longer daylight hours during this period. In contrast, a study on bunching onion cultivation in Poland reported the highest chlorophyll levels after a 60-day growing period [5]. However, in the present study, chlorophyll content was highest after a 90-day period. The extended growing time and the high temperatures, coupled with the intense light during July and August, likely contributed to chlorophyll degradation, making its accumulation more challenging [32,33]. Regarding varietal characteristics, KAMI, NATS, and NAKA were grouped together based on lower pigment compound content, while YAM1–YAM5 and YSG1 exhibited higher pigment compound levels (Appendix A). In terms of functional components, a difference was observed between NATS and YSG1; however, no significant differences were found among the other varieties (Appendix A). These findings clarify the varietal characteristics under different growing conditions in summer.

### 3.2. Estimation of the Metabolites Involved in Leaf Tipburn

Leaf tipburn is generally attributed to calcium deficiency. It has been reported that Ca^2+^, which moves along with water, tends to concentrate in mature leaves with active photosynthesis and transpiration, making young leaves more susceptible to Ca^2+^ deficiency [7]. In the case of Chinese chives (*A. tuberosum*), it has been suggested that dehydration is the cause, particularly in the lower leaves with high stomatal density at the leaf tips [34]. In this experiment, leaf tipburn was also observed in the lower leaves, suggesting a potential link with water stress. The cultivation experiment was conducted using heat-tolerant variety under various growth conditions. However, ANOVA analysis revealed no significant differences in leaf tipburn rates either between cultivars or among growth conditions (*p* > 0.05, see Appendix A). Nonetheless, variations in the incidence of leaf tipburn among individual plants suggest that multiple factors contribute to its occurrence.

In order to explore the underlying mechanisms, we conducted metabolome profiling of plants with high and low rates of leaf tipburn and analyzed the relationship between metabolites and leaf tipburn.The results from VIP scores based on PLS-DA (Figure 5B), volcano plots (Figure 6), and SHAP values (Figure 7) suggest a potential association between leaf tipburn and both flavonoids and organosulfur compounds. Flavonoids are one of the defensive antioxidant substances that play a critical role in plant stress responses [35]. They contribute to alleviating oxidative stress and protecting cell membranes, potentially influencing the occurrence of leaf tipburn through metabolic changes under stress conditions. Organosulfur compounds are involved in various stress responses, including the scavenging of reactive oxygen species (ROS) by glutathione, the synthesis of ethylene from methionine, and signal transduction via hydrogen sulfide ($H_{2}$S). Notably, $H_{2}$S plays a key role in inducing stomatal closure, promoting nitric oxide production, and enhancing ABA synthesis [36,37,38]. Comparative analyses of organosulfur compounds between the control and leaf tipburn groups revealed significant differences in key metabolic pathways. In the control group, metabolites associated with sulfur assimilation, glutathione, and methionine biosynthesis were increased, while gamma-Glu-PRENCSO, a precursor of pungency, was decreased (Figure 8). Gamma-Glu-PRENCSO is converted into PRENCSO, a precursor of pungency, by the action of gamma-glutamyl transferase (GGT), which cleaves the gamma-Glu group [39]. GGT plays a critical role in the synthesis of glutathione, which is involved in scavenging reactive oxygen species, by transferring gamma-Glu groups to specific amino acids or peptides. In Arabidopsis, GGT has been reported to be essential for mitigating oxidative stress [40]. Oxidative stress is caused by various factors, including environmental (abiotic) stress [41], and the ability to appropriately respond to such stress has been suggested to influence the occurrence of leaf tipburn. These findings indicate that stress-tolerant metabolic processes were more active in the control group, whereas these processes were diminished in the leaf tipburn group.

Regarding plant hormones, no significant differences in salicylic acid or jasmonic acid (JA), which are typically involved in plant defense responses, were observed between the two groups. However, the lack of increased levels of these hormones in the leaf tipburn group suggests insufficient activation of stress-related biological pathways [42,43]. Moreover, reductions in the ethylene precursor 1-aminocyclopropane-1-carboxylic acid and the ABA in the leaf tipburn group imply a diminished hormonal response to water stress, potentially impairing the plants’ ability to adapt to adverse environmental conditions. These findings suggest that, in the control group, organosulfur compounds were efficiently processed through metabolic pathways, enabling preparations for stress adaptation and mitigation. In the leaf tipburn group, however, organosulfur compounds may have been diverted toward the synthesis of pungency precursors, resulting in a metabolic bias that effectively compromised the plants’ response to water stress. These insights underscore the critical role of sulfur metabolism in regulating hormonal responses and highlight potential mechanisms contributing to leaf tipburn.

These results suggest that enhancing flavonoid accumulation and activating GGT to reduce gamma-Glu-PRENCSO could be critical strategies for mitigating oxidative stress and suppressing leaf tipburn. Our findings highlight the dual role of flavonoids as antioxidants and signaling molecules that can modulate plant responses to environmental stress and leaf tipburn symptoms, providing a biochemical foundation for developing stress-tolerant varieties. The activation of GGT to decrease gamma-Glu-PRENCSO levels could also serve as a metabolic intervention to reduce the buildup of reactive oxygen species, thereby preventing cellular damage and suppressing leaf tipburn. This insight could be applied in practical breeding programs to develop stress-resistant cultivars by targeting metabolic pathways associated with sulfur metabolism and flavonoid biosynthesis. Furthermore, future studies could explore the feasibility of using gamma-Glu-PRENCSO and flavonoids as biomarkers to monitor stress levels and predict leaf tipburn incidence in agricultural practices. Additionally, future research could explore the genetic and environmental factors influencing these metabolic pathways, as well as their interactions with other stress response mechanisms, to develop comprehensive strategies for improving crop resilience.

## 4. Materials and Methods

### 4.1. Materials and Growth Conditions

In this study, we examined the following: two $F_{1}$ cultivars from Nakahara Seed Product Co., Ltd. (Yokosuka, Japan)—‘Natsuhiko’ (NATS) and ‘Kaminari’ (KAMI); one Yamaguchi Prefecture purebred variety—‘YSG1go’ (YSG1); and six Yamaguchi Prefecture-bred $F_{1}$ hybrid lines—‘Yamakou01’ (YAM1), ‘Yamakou02’ (YAM2), ‘Yamakou03’ (YAM3), ‘Yamakou04’ (YAM4), ‘Yamakou05’ (YAM5), and ‘Nakayamakou’ (NAKA). These plant materials were grown in the greenhouse of Yamaguchi Prefectural Agriculture and Forestry General Technology Center (34° N, 131° E). We established six distinct growth conditions as follows: (1) sown on 7 May 2019 and harvested on 18 July 2019; (2) sown on 7 May 2019 and harvested on 22 August 2019; (3) sown on 7 May 2019 and harvested on 18 September 2019; (4) sown on 9 June 2019 and harvested on 22 August 2019; (5) sown on 9 June 2019 and harvested on 18 September 2019; and (6) sown on 4 July 2019 and harvested on 18 September 2019. Under growth conditions (1), (2), and (3), we cultivated eight varieties except YSG1, while under growth conditions (4), (5), and (6), we cultivated all nine varieties. This experiment aimed to screen for the conditions and cultivars most susceptible to leaf tipburn. To make the most of the limited greenhouse space and resources, we selected nine cultivars with three replications and six growth conditions in order to reflect diverse genetic and environmental scenarios.

The crops were planted in 6 rows on a 90 cm-wide ridge, with an inter-row spacing of 12 cm and a seeding density of 120 seeds per square meter. A total of 1.0 kg of nitrogen per area (kg/a) was applied during the experiment. This nitrogen was divided into two applications: an initial 0.5 kg/a as a basal fertilizer followed by an additional 0.5 kg/a as a topdressing later in the two-leaf stage. The watering conditions were as follows: on the day of sowing, water was applied at 48 L/$m^{2}$, with a soil water tension value set to 1.5 pF. During germination (0–4 days), irrigation was provided at 24 L/$m^{2}$ per application, maintaining a pF value between 1.5 and 1.6. In the cotyledon stage (4–11 days), 6 L/$m^{2}$ of water was applied every 2–3 days, keeping the pF value between 1.6 and 1.8. At the one-leaf stage (11–18 days), 10 L/$m^{2}$ of water was supplied every 2–3 days, maintaining a pF value between 1.7 and 2.0. During the two-leaf stage (18–28 days), irrigation was increased to 12 L/$m^{2}$ daily, with the pF value maintained between 1.6 and 1.8. During the three-leaf stage (28–38 days), irrigation was performed every 3 days with 6 L/$m^{2}$ to maintain a pF value between 2.0 and 2.3. In the four-leaf stage (38–49 days), water was applied almost daily at 12 L/$m^{2}$, maintaining a pF value of 1.8–2.0. During the five-leaf stage (49–59 days), irrigation was carried out every 3 days with 5 L/$m^{2}$, keeping the pF value between 2.0 and 2.5. At the six-leaf stage (59 days onward), irrigation was reduced to 0–1 time per week with 4 L/$m^{2}$, maintaining a pF value of 2.3–2.5 until harvest. After harvest, the plants were grouped into bundles of 10, with three bundles used for biological replicates (*n* = 3). The outer leaves were removed, leaving two young leaves for further analysis.

Meteorological data, including air temperature (average, maximum, and minimum), relative humidity (average and minimum), and sunshine duration for Yamaguchi City (34° N, 131° E), as well as solar radiation data for Fukuoka City (33° N, 130° E), the nearest observation site to Yamaguchi City, were obtained from the Japan Meteorological Agency for the period from May to September 2019 (https://www.data.jma.go.jp/obd/stats/etrn/, (accessed on 30 December 2024)) (Appendix A).

### 4.2. Leaf Tipburn Measurement

The samples were arranged on a black background to ensure that the leaves did not overlap, and images were captured. From the obtained images, pixel counts for leaf tipburn areas and healthy (green color) areas were calculated using the free software LIA32 Ver. 0.376β1, which has a supervised classification function (https://www.agr.nagoya-u.ac.jp/~shinkan/LIA32/download.html, accessed on 30 December 2024) [44]. The training images included ‘Asagi-kujyo harvested on 18 August 2017’ (Appendix A), with 50 training points for each class: leaf tipburn area, healthy area, and background. The percentage of leaf tipburn was calculated as follows: (pixel count of leaf tipburn area/pixel count of total plant area) × 100.

### 4.3. SPAD Value Measurement

The SPAD value was determined by averaging three measurements taken at the center of the leaf using a Soil Plant Analysis Development chlorophyll meter (SPAD-502Plus, KONICA MINOLTA, Inc., Tokyo, Japan) [45]. For each sample, the overall SPAD value was calculated by averaging the SPAD readings from the first (youngest) and second (second youngest) leaves.

### 4.4. Pigment Compound Measurement

The method for measuring pigment compound was developed with reference to Dissanayake et al. [46]. Chlorophyll *a*, Chlorophyll *b*, Lutein, β-carotene, violaxanthin, and neoxanthin were measured using high-performance liquid chromatography (HPLC L-7000 series, HITACHI, Tokyo, Japan). The leaf blade of the trimmed sample was cut into approximately 5 mm sections using a kitchen knife, and a portion was weighed to precisely 5.0 g for analysis. The sample was placed into a homogenizer cup, followed by the addition of 0.2 g of basic magnesium carbonate, 0.1 g of butylated hydroxytoluene, and 20 mL of cold 100% acetone. The premix was homogenized at 10,000 rpm for 5 min using an Ace HOMOGENIZER AM-7 (Nihon Seiki Co., Ltd., Tokyo, Japan), with the outer chamber of the cup filled with ice to keep the cup cool during the process. The suspension was filtered using suction filtration, and the filtrate was collected. The residue was transferred to a mortar placed on ice, mixed with 10 mL of cold 100% acetone, and ground with a pestle. The resulting mixture was filtered again using suction filtration to collect the filtrate. This process was repeated three times in order to maximize filtrate recovery. The combined filtrates were dried by adding 5 g of anhydrous sodium sulfate and were left in the dark for 30 min. Subsequently, the mixture was passed through a filter (Advantec, Tokyo, Japan) to quantify the volume. Finally, the filtrate was passed through a 0.45 μm filter (Advantec, Tokyo, Japan) and was used as the extract for HPLC analysis. The extract was analyzed using HPLC equipped with a UV–VIS detector (HITACHI L7420) set at 435 nm for carotenoid detection. Pigmented compounds were separated using a LiChroCART 250-4.0 Lichrospher 100 RP-18 5 μm column (KANTO CHEMICAL, Tokyo, Japan). The separation was performed using gradient elution with two solvents: (A) 80% methanol solution, prepared by mixing 400 mL of HPLC-grade methanol, 50 mL of ultra-pure water, and 50 mL of 100 μM HEPES buffer (pH 7.5), and (B) a 50:50 mixture of ethyl acetate and 80% methanol. The gradient elution program was as follows: it started with 100% solvent A for 20 min; then, a linear gradient was followed to solvent B over 50 min. The flow rate was maintained at 1 mL/min, with a column temperature of 30 °C and an injection volume of 20 μL.

### 4.5. 70% Ethanol Extraction

The method for extraction was developed with reference to Hang et al. [47]. The leaf blades of the trimmed sample were cut with a kitchen knife at intervals of approximately 5 mm and weighed out to 5.0 g. The tissue samples were placed in a conical flask containing 35 mL of ethanol and 10 mL of distilled water (final concentration 70%) and were boiled at 80 °C in a water bath for 15 min. The mixture was then homogenized at 10,000 rpm for 5 min using an Ace HOMOGENIZER AM-7 (Nihon Seiki Co., Ltd., Tokyo, Japan), with the outer chamber of the cup filled with ice to keep the cup cool during the process. The resulting suspension was filtered using suction filtration, and the filtrate was collected and quantified. The extract was stored at −20 °C in a freezer for further analysis.

### 4.6. Total Phenolic Compounds Measurement

The total phenolic compounds content in the 70% ethanol extract was measured using the Folin–Ciocalteu method [48]. The 70% ethanol extraction was diluted 5 times with distilled water. Of the diluted solution, 1 μL was put into test tube, and 1 mL of 1 N Folin–Ciocalteu reagent was added. After 3 min, 1 mL of 10% sodium carbonate solution was added to the test tube. The mixture was incubated for 1 h at room temperature in the dark. The absorbance of the reaction mixture was measured at 530 nm with a spectrophotometer (U-2000, Hitachi High-Tech corporation, Tokyo, Japan). Catechol was used as a standard, and the results were expressed as milligrams of catechol equivalent (mg CA) per 100 g fresh weight of plant sample.

### 4.7. Total Flavonoid Compound Measurement

The total flavonoid compound content in the 70% ethanol extract was measured using the method described previously [49]. The 70% ethanol extract and n-hexane were put into a test tube with a screw cap at a ratio of 1:1 and were stirred to dissolve the chlorophyll, carotenoids, and other pigments into the hexane layer. The ethanol layer was diluted 2 times with 70% ethanol. Three mL of the diluted solution was put into a test tube, and 9 mL of 2% aluminum chloride solution was added. The mixture was incubated for 1 h at room temperature in the dark. The absorbance of the reaction mixture was measured at 420 nm with a spectrophotometer (U-2000, Hitachi High-Tech Corporation, Tokyo, Japan). Quercetin was used as a standard, and the results were expressed as milligrams of quercetin equivalent (mg quercetin) per 100 g fresh weight of the plant sample.

### 4.8. Fructan Measurement

The fructan content in the 70% ethanol extract was measured using the thiobarbituric acid method [50], with minor modifications. The 70% ethanol extraction was diluted 5 times with 70% ethanol. Of the diluted solution, 20 μL was put into a test tube, and 10 μL of 25 mM ammonium acetate buffer was added. Furthermore, 10 μL of invertase solution was added and left at room temperature for 5 min to hydrolyze the sucrose in the extract. Measures of 50 μL of distilled water and 10 μL of 10 N sodium hydroxide solution were added and heated in boiling water for 10 min to decompose the fructose in the solution. After cooling rapidly in ice water, 1 mL of thiobarbituric acid solution and 1 mL of 12 N hydrochloric acid were added and heated in boiling water for 6 min. The absorbance of the reaction mixture was measured at 432 nm with a spectrophotometer (U-2000, Hitachi High-Tech Corporation, Tokyo, Japan). Fructan was used as a standard, and the results were expressed as milligrams of quercetin equivalent (mg 1-kestose) per 100 g fresh weight of the plant sample.

### 4.9. Metabolome Analysis

Sample preparation was performed automatically by a liquid-handling system (Microlab STAR Plus, Hamilton, Company, Reno, NV, USA) for dispensing, plate transfer, solvent drying (Ultravap Mistral, Porvair PLC, Norfolk, UK), dissolving, and filtration, as described by Sawada et al. [51]. The leaf blades of the trimmed samples were frozen in liquid nitrogen. The frozen samples were dried for three days using a freeze-dryer (TAITEC VD-250R freeze-dryer coupled with a vacuum pump, TAITEC, Saitama, Japan) and then powdered using a mill (KMZ-0401/P, Koizumi Seiki Corporation, Osaka, Japan). The powders were weighed out to 4 mg in a 2 mL tube, and a 5 mm zirconia ball (NIKKATO CORPORATION, Osaka, Japan) was added. One milliliter of extraction solvent (80% methanol and 0.1% formic acid coupled with 8.4 nmol/L lidocaine and 210 nmol/L 10-camphorsulfonic acid as internal standard) was added to a 2 mL tube (resulting in a concentration of 4 mg/mL), and the metabolites were extracted in a multi-bead shaker (Shake Master NEO, Biomedical Science, Tokyo, Japan) at 1000 rpm for 2 min. After centrifugation at 9100 G for 1 min, the supernatants were diluted 4 times with an extraction solvent (resulting in a concentration of 1 mg/mL). Of each diluted solution, 25 μL was transferred to a 96-well plate, dried with nitrogen air, redissolved in 250 μL of ultra-pure water (LC–MS grade), and filtered using a 0.45 μm pore size filter plate 384 (Multiscreen HTS 384-Well HV, Merck, Rahway, NJ, USA). One microliter of the solution extract at a final concentration of 100 ng/μL was subjected to widely targeted metabolomics using LC-QqQ-MS (UHPLC–Nexera MP/LCMS-8050, SHIMADZU, Tokyo, Japan). The solution extract was separated using a ACQUITY UPLC HSS T3 Column, 100 Å, 1.8 µm, 1 mm × 50 mm (Waters Corp., Milford, CT, USA). The separation was performed using gradient elution with two solvents: (A) 0.1% (*v*/*v*) formic acid in distilled water, and (B) 0.1% (*v*/*v*) formic acid in acetonitrile. The flow rate was maintained at 0.24 mL/min, with a column temperature of 30 °C. The setting parameters for LC–QqQ-MS analysis are summarized in the Appendix A.

### 4.10. Data Analysis

#### 4.10.1. Statistical Analysis

Data for the SPAD values, pigment compounds (chlorophyll *a*, chlorophyll *b*, lutein, β-carotene, violaxanthin, neoxanthin), and functional components (total phenolic compounds, total flavonoid compounds, fructan), along with leaf tipburn rate, were analyzed using one-way analysis of variance (ANOVA) and Tukey’s multiple comparison test for each individual growing condition and across all conditions. A significance level of *p* < 0.05 was applied. An analysis of the secondary metabolite characteristics of the varieties was conducted. The dataset, including chlorophyll *a*, chlorophyll *b*, lutein, β-carotene, violaxanthin, neoxanthin, total phenolic compounds, total flavonoid compounds, and fructan, was first standardized using the StandardScaler function from Scikit-Learn. Subsequently, the standardized data were analyzed using principal component analysis (PCA) with two components (n_components = 2) and K-means clustering (n_clusters = 4, random_state = 42) to assess the variation in metabolite profiles across samples. The analysis was performed using the Python 3.9.7 with Pandas 1.3.4, Scikit-Learn 0.24.2, and Matplotlib 3.4.3 libraries.

#### 4.10.2. Estimate Compounds Associated with Leaf Tipburn Using Metabolomics

A total of 472 metabolite intensities were obtained from the metabolome analysis. Missing values were imputed with a value of 10, and the signal intensities for all samples (*n* = 3) were averaged. Metabolites with signal-to-noise ratios (S/N, defined as the ratio of the average signal intensity to that of the extraction solvent control) of less than 5 across all experimental groups were excluded. Additionally, metabolites with relative standard deviations (RSD) greater than 0.3 in all experimental groups were removed, resulting in a final dataset of 267 metabolites. The data matrix was then normalized to the median and auto-scaled. The processed data were used for comparative analysis (Appendix A). Initially, partial least squares discriminant analysis (PLS-DA) was performed on all 267 metabolites, and VIP scores were used to assess the overall trends in metabolomic data between the leaf tipburn group (top 5% in leaf tipburn rate) and the control group (bottom 5% in leaf tipburn rate). This method is particularly suitable for high-dimensional metabolomic data as it emphasizes group separation and identifies key metabolites using VIP scores. However, despite limitations such as assumptions of linearity and the risk of overfitting, it remains a valuable tool for exploring group separation in high-dimensional datasets. Additionally, a Student’s *t*-test, followed by false discovery rate (FDR) correction using the Benjamini–Hochberg procedure, was conducted to compare metabolites between groups, identifying those with significant differences (FDR < 0.05) and a change of 1.5-fold or more ($\log_{2}FC$ [Leaf tipburn/control] ≥ 0.58 or ≤−0.58). These metabolites were then used to create volcano plots. The preprocessing steps, including missing value imputation, S/N filtering, RSD calculations, and intensity correction via dividing by internal standards, were performed using Python 3.9.7 with Pandas 1.3.4. Data normalization and subsequent analysis were conducted using MetaboAnalyst 6.0.

#### 4.10.3. Machine Learning Methods

A random forest regression model was employed to predict the leaf tipburn rate using 267 metabolite profiles as input features. This method was chosen to address the limitations of PLS-DA, specifically its inability to capture non-linear relationships, and to mitigate the risk of overfitting inherent in single decision tree models. The data, which included all available samples, were first split into training and test sets using a 70–30 split, ensuring a consistent random state = 42 to enable reproducibility. The training set was used to fit the random forest model, which was optimized based on the following hyperparameters: n_estimators = 100, max_features = ‘log2’, min_samples_leaf = 2, min_samples_split = 2, and max_depth = none. To gain insights into the model’s predictions, Shapley additive explanations (SHAP) analysis was performed. All analyses and visualizations were conducted using Python 3.9.7. The following libraries were used: Pandas 1.3.4, Scikit-Learn 0.24.2, SHAP 0.46.9, and Matplotlib 3.4.3.

## 5. Conclusions

In this study, metabolome profiling was conducted to compare leaf tipburn and control groups, revealing that organic sulfur compounds are involved in leaf tipburn. Among these, gamma-Glu-PRENCSO was found to significantly increase in the leaf tipburn group. Gamma-Glu-PRENCSO is an intermediate of PRENCSO, a precursor of pungency, and is synthesized when GGT removes the gamma-glutamyl group. This gamma-glutamyl group functions as a substrate for glutathione, which plays a crucial role in stress responses, reactive oxygen species scavenging, and the recycling of sulfur compounds. These findings suggest that activating GGT to reduce gamma-Glu-PRENCSO accumulation could be a promising strategy to enhance stress tolerance and mitigate leaf tipburn. Future research will aim to monitor the dynamics of gamma-Glu-PRENCSO under various conditions to further elucidate the mechanisms underlying leaf tipburn. Additionally, based on these insights, efforts will be made to establish optimal cultivation conditions and develop stress-tolerant cultivars.

## Figures and Tables

**Figure 1 plants-14-00187-f001:**
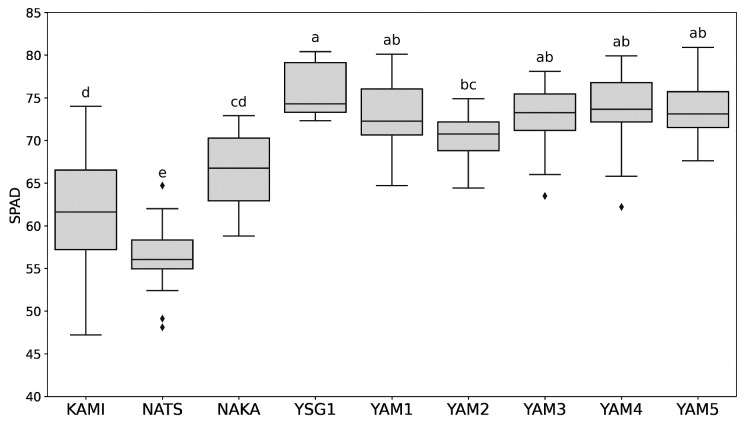
Boxplot of SPAD values by varieties and lines under all growing conditions. The box represents the interquartile range, the horizontal line within the box indicates the median, and diamonds represent outliers. Different letters indicate significant differences at *p* < 0.05.

**Figure 2 plants-14-00187-f002:**
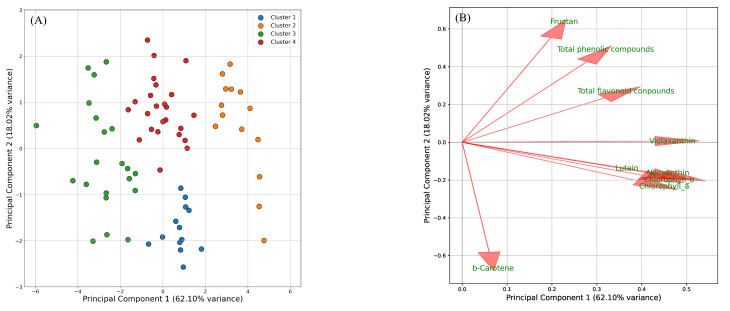
Principal component analysis (PCA) score plot with K-means clustering (**A**) and PCA loading plot (**B**) of the dataset from May sowing and July, August, and September harvests. (**A**) Samples are color-coded on K-means clustering, with four clusters having been identified. (**B**) Arrows represent the direction and magnitude of each variable’s influence, with longer arrows indicating stronger contributions.

**Figure 3 plants-14-00187-f003:**
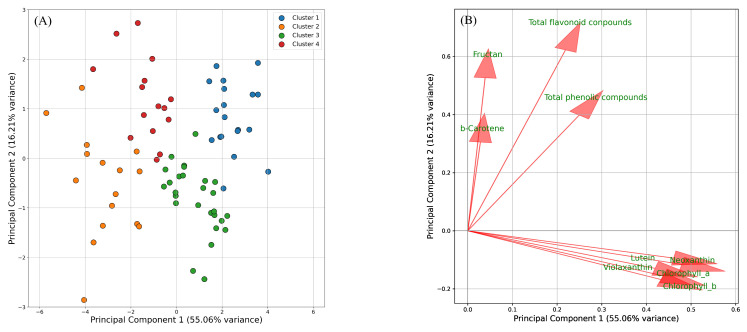
Principal component analysis (PCA) score plot with K-means clustering (**A**) and PCA loading plot (**B**) of a dataset from the May, June, and July sowing with September harvest. (**A**) Samples are color-coded on K-means clustering, with four clusters identified. (**B**) Arrows represent the direction and magnitude of each variable’s influence, with longer arrows indicating stronger contributions.

**Figure 4 plants-14-00187-f004:**
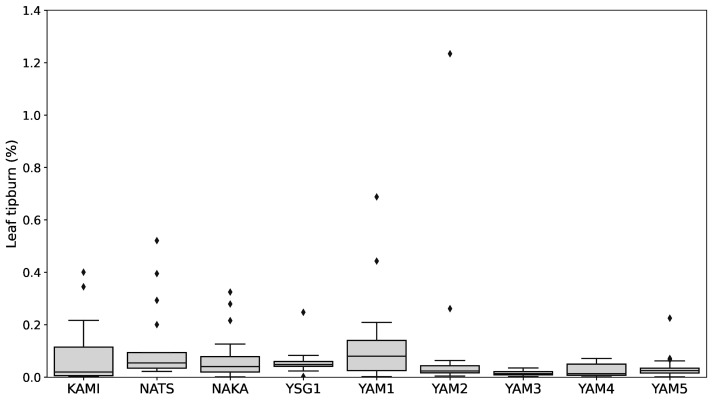
Boxplot of leaf tipburn by varieties and lines under all growing conditions. The box represents the interquartile range, the horizontal line within the box indicates the median, and diamonds represent outliers.

**Figure 5 plants-14-00187-f005:**
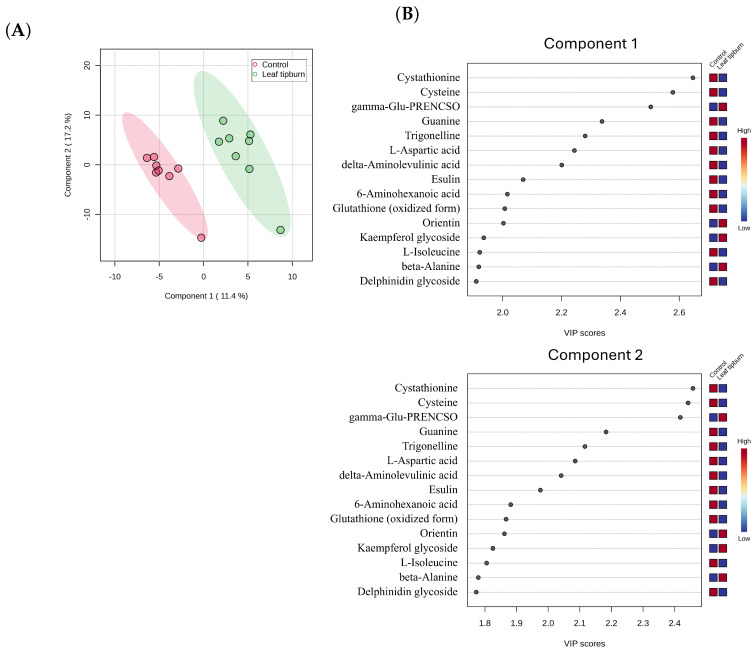
(**A**) Partial least squares discriminant analysis (PLS-DA) scores plot and (**B**) VIP (variable importance in projection) score plot of the metabolite profiles for the samples with the highest and lowest leaf tipburn rates. The contributions of the metabolites to the Component 1 and Component 2 axes are color coded, based on the contribution scale according to the VIP score. gamma-Glu-PRENCSO: gamma-glutamyl-propenyl cysteine sulfoxide.

**Figure 6 plants-14-00187-f006:**
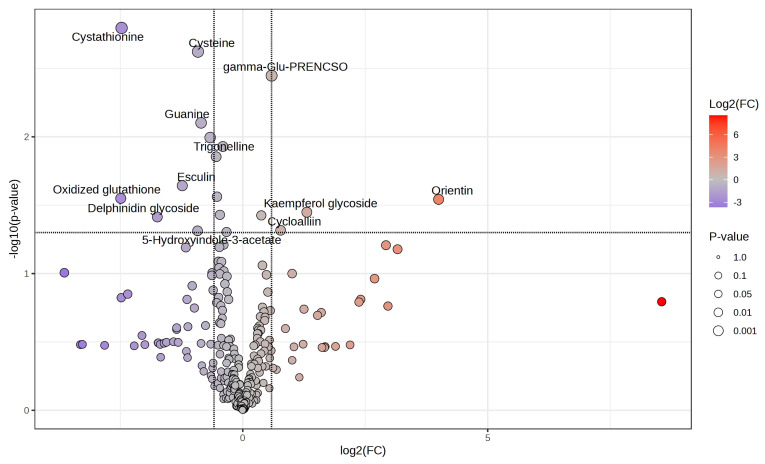
Volcano plot of the metabolite profiles for the highest and lowest leaf tipburn rates. The x-axis shows the $\log_{2}FC$-fold changes (leaf tipburn group/control group). The y-axis represents the −$\log_{10}FC$-transformed *p*-values. Each point corresponds to a metabolite, with its position indicating both the magnitude of change and the statistical significance. Points on the right side (positive $\log_{2}FC$-fold change) indicate metabolites elevated in the leaf tipburn group, while points on the left side (negative $\log_{2}FC$-fold change) indicate metabolites elevated in the control group. Metabolites with significant *p*-values (*p* < 0.05) are highlighted, with circle size corresponding to the significance level and color representing the $\log_{2}FC$-fold change magnitude (red—higher in the leaf tipburn group; purple—higher in the control group). gamma-Glu-PRENCSO: gamma-glutamyl-propenyl cysteine sulfoxide.

**Figure 7 plants-14-00187-f007:**
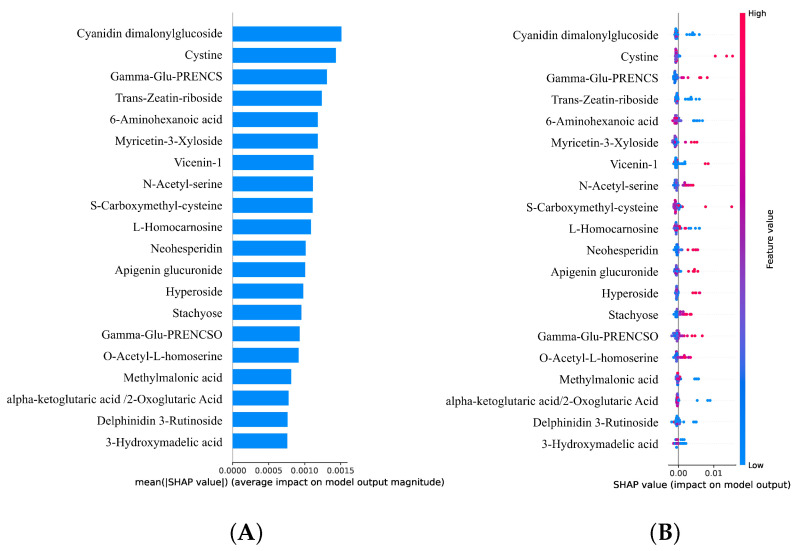
Shapley additive explanations (SHAP) analysis of metabolite profile influence on the leaf tipburn rate. (**A**) The bar plot of the SHAP mean value; (**B**) the summary plot of the top 20 most influential features. The feature ranking on the y-axis represents the order of importance for each feature in the prediction model. SHAP values on the x-axis indicate the predictive power of the model. Each row is plotted with dots representing the influence on individual validation data points. Red dots (high values) indicate a stronger prediction of high leaf tipburn rate, while blue dots (low values) indicate a stronger prediction of low leaf tipburn rate. In the summary plot, red dots indicate high metabolite concentrations, while blue dots indicate low concentrations. Dots on the positive side of the x-axis represent metabolites that contribute to an increase in leaf tipburn rate. Example interpretation: On the positive side of the x-axis, red dots indicate that higher concentrations of gamma-Glu-PRENCSO are associated with an increased leaf tipburn rate. gamma-Glu-PRENCSO—gamma-glutamyl-propenyl cysteine sulfoxide; gamma-Glu-PRENCS—gamma-glutamyl-propenyl cysteine.

**Figure 8 plants-14-00187-f008:**
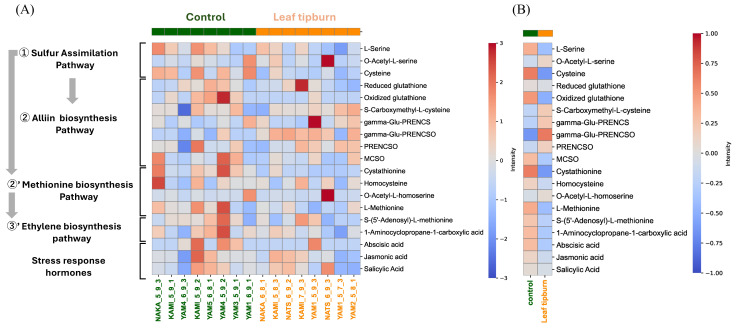
Comparison of the leaf tipburn group and control group based on organosulfur compounds and stress response hormones. (**A**) Heat map comparing the metabolite profiles between the leaf tipburn group (top 5% rate of leaf tipburn) and the control group (bottom 5% rate of leaf tipburn). Each column represents an individual sample. (**B**) Heat map showing the average values of each metabolite for the two groups. Each column represents an individual sample, and the rows indicate specific metabolites categorized into sulfur assimilation, alliin biosynthesis, methionine biosynthesis, ethylene biosynthesis pathways, and stress response hormones. gamma-Glu-PRENCS—gamma-glutamyl-propenyl cysteine; gamma-Glu-PRENCSO—gamma-glutamyl-propenyl cysteine sulfoxide; PRENCSO—propenyl cysteine sulfoxide; MCSO—methyl cysteine sulfoxide.

**Table 1 plants-14-00187-t001:** Sample distribution by varieties or lines and growth conditions based on K-means clustering of a dataset from May sowing and July, August, and September harvests. The table summarizes the assignment of samples to four distinct clusters (cluster 1 to cluster 4) identified through K-means clustering.

Cluster	Features	Growth Conditions	KAMI	NATS	NAKA	YAM1	YAM2	YAM3	YAM4	YAM5
cluster 1	β-carotene_UP	May_July ^y^			1	3	1	2	1	3
		May_August				2				
		May_September								
cluster 2	Pigment cmpds_UP ^z^	May_July								
		May_August								
		May_September				3	3	3	2	3
cluster 3	Pigment cmpds_DOWN	May_July	3	3	2		2		2	
		May_August	3	3	2					
		May_September		1						
cluster 4	Functional cmpds_UP	May_July						1		
		May_August			1	1	3	3	3	3
		May_September	3	2	3				1	

^y^: Indicates the sowing month and the harvesting month. ^z^: cmpds stands for compounds.

**Table 2 plants-14-00187-t002:** Sample distribution by varieties or lines and growth conditions based on the K-means clustering of a dataset from May, June, and July sowing and September harvest. The table summarizes the assignment of samples into four distinct clusters (cluster 1 to cluster 4) identified through K-means clustering.

Cluster	Features	Growth Conditions	KAMI	NATS	NAKA	YSG1	YAM1	YAM2	YAM3	YAM4	YAM5
cluster 1	Pigment cmpds_UP ^y^ Functional cmpds_UP	May_September ^z^					3	3	2	2	2
		June_September					2				1
		July_September				3					1
cluster 2	Pigment cmpds_DOWN Functional cmpds_DOWN	May_September		2	1						
		June_September	3	2							1
		July_September	3	3	1						
cluster 3	Pigment cmpds_UP Functional cmpds_DOWN	May_September							1		1
		June_September			1	2	1	1	3	3	
		July_September			1		3	3	3	3	2
cluster 4	Pigment cmpds_DOWN Functional cmpds_UP	May_September	3	1	2					1	
		June_September		1	2	1		2			1
		July_September			1						

^y^: cmpds stands for compounds. ^z^: Indicates the sowing month and the harvesting month.

## Data Availability

The data used in the study were obtained from the websites of the Japan Meteorological Agency (https://www.data.jma.go.jp/obd/stats/etrn/, accessed on 30 December 2024) and the raw MS data can be downloaded from Drop Met database (https://prime.psc.riken.jp/menta.cgi/prime/drop_index#DM0065, (accessed on 11 November 2024)).

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
